# Entelon (*vitis vinifera* seed extract) reduces degenerative changes in bovine pericardium valve leaflet in a dog intravascular implant model

**Gab-Chol Choi[1,2‡], Sokho Kim[3‡], Md. Mahbubur Rahman[3], Ji Hyun Oh[4], Yun Seok Cho[4], Hong Ju Shin[4]** *

**1** Animal Medical Center W, Mapo-gu, Seoul, Korea, **2** Department of Veterinary Surgery, College of Veterinary Medicine, Jeonbuk National University, Jeonju-si, Jeollabuk-do, Republic of Korea, **3** KNOTUS Co., Ltd., Research Center, Incheon, Korea, **4** Department of Thoracic and Cardiovascular Surgery, Korea University Ansan Hospital, Korea University College of Medicine, Ansan, Korea

‡ These authors are contributed equally as co-first authors.
* babymedi@naver.com

**Data Availability Statement:** All relevant data are within the manuscript and its Supporting Information files.

## Abstract

### Background and aims

Inflammation and calcification are major factors responsible for degeneration of bioprosthetic valve and other substitute heart valve implantations. The objective of this study was to evaluate the anti-inflammatory and anti-calcification effects of Entelon150® (consisting of grape-seed extract) in a beagle dog model of intravascular bovine pericardium implantation.

### Methods

In total, 8 healthy male beagle dogs were implanted with a bovine pericardium bilaterally in the external jugular veins and divided into two groups. Animals in the Entelon150® group (n = 4) were treated with 150 mg of Entelon150® twice daily for six weeks after surgery. The negative control (NC) group (n = 4) was treated with 5 ml of saline using the same method. After six weeks, we measured the calcium content, performed histological examination, and performed molecular analysis.

### Results

The calcium content of implanted tissue in the Entelon150® group (0.56±0.14 mg/g) was significantly lower than that in the NC group (1.48±0.57 mg/g) (p < 0.05). Histopathological examination showed that infiltration of chronic inflammatory cells, such as fibroblasts and macrophages, occurred around the graft in all groups; however, the inflammation level of the implanted tissue in the Entelon150® group was s lower than that in the NC group. Both immunohistochemical and western blot analyses revealed that bone morphogenetic protein 2 expression was significantly attenuated in the Entelon150® group.

**Funding:** This study was supported by a Korea University Grant (K1924931).

**Competing interests:** The authors report no conflicts of interest including Sokho Kim and Md. Mahbubur Rahman. It is noted that Sokho Kim and Md. Mahbubur Rahmanis employed by KNOTUS Co., Ltd. This does not alter any adherence to PLOS ONE policies on sharing data and materials.

## Conclusions

Our results indicate that Entelon150® significantly attenuates post-implantation inflammation and degenerative calcification of the bovine pericardium in dogs. Therefore, Entelon150® may increase the longevity of the bovine pericardium after intravascular implantation.

## Introduction

Bovine pericardium is widely used in patch material during vascular surgery. It is also used in bioprosthetic heart valve leaflets with specific treatment to increase its longevity. Heart valves made of a bovine pericardium are safe, offer improved hemodynamics, have less risk of thrombosis, and do not need long-term anticoagulant therapy [1, 2]. However, the durability of the bovine pericardium is a major problem, as it is prone to valve calcification, structural deterioration, and eventual failure. Several approaches to reduce calcification have been attempted, including systemic anti-calcification agent administration. However, many of these approaches have been either ineffective or have produced unwanted side effects [3]. To overcome this drawback, studies on the complementary treatment of calcification have focused on traditional herbal medicines recently [4–6].

Grape fruit (*Vitis vinifera*) is one of the most important and popular fruit crops worldwide because of its high phytochemical content and its nutritional value. All parts of this fruit have been used as dietary supplements to treat or prevent various diseases [7–9]. In general, grape fruit is rich in phenols, flavonoids, and fatty acids. The anti-inflammatory [7, 9] and anti-oxidative [7] effects of grape seed extract (GSE) are now well established.

Interleukin-6 (IL-6) plays an important role in increasing bone morphogenic protein 2/4 (BMP-2/4) expression in vessels and valve tissue, thereby leading to vascular calcification [3]. It has been shown that application of GSE reduces IL-6 activity in different disease models [10, 11].

To the best of our knowledge, there have been no studies examining the effects of GSE on bovine pericardium implants. It is thus unknown whether GSE could prevent bovine pericardium calcification and degeneration. Therefore, the purpose of this study was to investigate the anti-inflammatory and anti-calcification effects of Entelon150® on intravenous bovine pericardium implants in beagle dogs.

## Methods

### Animal and experimental design

Eight healthy male beagle dogs (20 weeks old; mean body weight 9.76 ± 0.32 kg, range 8.00–10.80 kg) were used in this study which were purchased from ORIENT BIO Inc. (Seongnam, Gyeonggi-do, Republic of Korea). They were housed separately in stainless steel cages (W 895 × L 795 × H 765 mm) in an environmentally controlled room (temperature 23 ± 3˚C, relative humidity 55 ± 15%, ventilation frequency 10–20 times/hr, light cycle 8 am to 8 pm, illumination 150 to 300 Lux). Food and sterilized water were available ad libitum. Animals were divided equally into two groups: 1) Negative control (NC) group: vehicle-treated after prosthetic implantation; 2) Entelon150®-treated group: treated with grape seed extract (Entelon150®; Lot number: RNR601, Hanlim Pharm. Co., Ltd. Yongin-si, Korea), 150 mg per animal twice daily for six weeks after implantation. The mouth of the animal was opened to its

natural position in the breeding box, and the test article was placed on the tongue. The animal's mouth was then shut, and the neck was gently stroked until the animal swallowed. All animals were closely monitored during the experimental period, and we observed no clinical symptoms. This study was approved by the Institutional Animal Care and Use Committee at the KNOTUS Co., Ltd., Incheon-si, Korea (Certificate No: IACUC 19-KE-132).

## Surgical procedure for implantation and postoperative care

Each animal was anesthetized with an intravenous injection of Zoletil 50 (VIRBAC, France; 5 mg/kg) and xylazine (Rompun®, Bayer AG, Germany; 2.5 mg/kg). After intubation with a 3.0-mm endotracheal tube, inhaled isoflurane was used to maintain anesthesia. All animals received 0.9% saline (10 ml/kg/h) intravenously throughout the surgical procedure. Intravenous antibiotic cephradine (30 mg/kg) and intravenous analgesic tramadol (2 mg/kg) were injected preoperatively as described previously [12]. The skin was incised along the ventral cervical midline, and blunt separation was performed to expose the left external jugular vein from the sternohyoid muscle. After administration of heparin (50 IU/kg, IV), the left jugular vein was temporarily blocked using 4–0 silk and 5 French feeding tubes. An 8-mm longitudinal incision was then made in the jugular vein using a no. 11 surgical blade. A commercially available bovine pericardium (PERIBORN® Bovine Pericardium, BP0506, Taewoong Medical Co., Ltd. Gimposi, South Korea) was used in this study. Before implantation, the Bovine pericardium was rinsed for 30 min in 500 ml of sterile physiological saline. An approximately 3-mm rectangular bioprosthetic was fixed to the inner wall of the jugular vein using a 6–0 polypropylene running suture. The jugular vein was closed by angioplasty using a 6–0 polypropylene running suture. The bovine pericardium was also implanted into the right external jugular vein as described above. The schematic diagram below shows the anatomy of the jugular vein in beagle dogs, relevant to this surgical procedure (Fig 1). Postoperatively, intramuscular enrofloxacine (10 mg/kg, bid), intravenous cephradine (30 mg/kg bid or tid), intravenous tramadol (2–3 mg/kg bid or tid) and intravenous cimetidine (10 mg/kg, bid) were performed for three days [12]. Respiratory rate was monitored by the movement of the abdominal and chest wall (breath/min) twice daily. Heart rate and ECG was recorded with a Cardiofax ECG-9020 electrocardiograph (Nihon Kohden, Tokyo, Japan) once daily for three days [13]. Beside these normal clinical signs were closely monitored daily until end of the experiment to identify any abnormality due to surgery or any side effect for administering Entelon150 including changes of body weight, feeding and drinking behavior, urination and defecation frequency, bleeding, salivating, vomiting, abnormality or redness of skin and eye color, abnormal sound and abnormal movement but none were observed in entire experimental period. Body weight was measured just before starting experiment and then once a week until six weeks (at o, 1, 2, 3, 4, 5 and 6 weeks) (Fig 2).

## Measurement of vascular patency

Vascular patency was checked using the color doppler mode of an ultrasonic device (LOGIQ e Ultrasound; GE Healthcare, Fairfield, USA) 6 weeks after the bioprosthetic implantation. Angiography was also performed with a bolus intravenous injection of 2 ml/kg iohexol (Omnipaque™, 300 mg I/ml; GE Healthcare) using a CT-scanner (CT; 16-channel multidetector; BrightSpeed Elite, GE Healthcare, Fairfield, CT, USA) for additional confirmation just after ultrasonography and immediately before sacrifice at 6 weeks.

## Sample collection

Six weeks after surgery and immediately after measuring vascular patency, the animals were euthanized by intravenous Propofol (10 mg/kg) and then intravenous Pentobarbital (50 mg/

**Fig 1. Anatomical diagram of a beagle's jugular vein during the implantation procedure.** Yellow star indicates the incision site of jugular vein.

kg). and both the left and right external jugular veins were removed. Only implanted prosthetic was collected from the right external jugular vein and connective tissues around the implanted prosthetic was carefully removed and then stored in a cryogenic freezer maintained at about

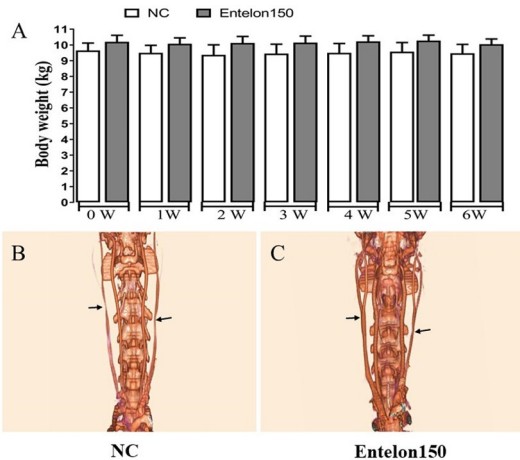

**Fig 2. Effects of bioprosthetic valve implantation and Entelon® treatment on body weight and vascular patency as measured by CT-Scan.** Arrow indicates the incision and implanted site in jugular vein.

-80˚C until calcium quantification. However, half of the left external jugular vein along with connective tissue was fixed in a 10% neutral buffered formalin solution for histopathological examination, and from the other half part only prosthetic was stored in a cryogenic freezer maintained at -80˚C for western blotting.

## Measurement of calcium content

Moisture was removed from the stored implant through 24 h of freeze-drying. After weighing the implants, they were suspended in a beaker containing 5 ml of aqua regia solution. Hydrogen peroxide ($H_2O_2$; 5 ml) was then added to the beaker, and the beaker was slowly heated to 70–80˚C on a hot plate for 6 h to dissolve the tissue. The solution was then heated for a further 5 h at 140–150˚C. An additional 5 ml of $H_2O_2$ was then added. Subsequently, each sample was diluted to a total volume of 50 ml with distilled water. The diluted samples were used to measure the calcium content.

## Histopathological examination

For histopathological examination, each sample was soaked in 10% neutralized buffered formalin and processed using standard methods. Paraffin-embedded tissue was sliced into 4-μm-thick sections and stained with hematoxylin and eosin (H & E). For H & E-stained slides, infiltration of inflammatory cells around the implanted bovine pericardium was quantified according to the following criteria: 0: none, 1: weak inflammatory cell infiltration, 2: moderate inflammatory cell infiltration, 3: severe inflammatory cell infiltration.

For immunohistochemical analysis, the sections were deparaffinized following standard protocols. The sections were then incubated in 3% hydrogen peroxide for 10 min to inactivate endogenous peroxidase and then blocked by incubating the sections for 1 h in 5% bovine serum albumin. The sections were then incubated overnight with anti-alpha smooth muscle actin (α-SMA) (1:1600, ab5964, Abcam) and bone morphogenetic protein 2 (BMP-2) (1:200, orb251474, Biorbyt) primary antibodies at 4˚C. The sections were washed and incubated with secondary anti-rabbit (α-SMA) or anti-mouse (BMP-2) antibodies for 30 min at approximately 20˚C. The sections were then washed, counterstained with Mayers Hematoxylin, and mounted on slides. The percentage of stained area (%) was compared between groups. For immunohistochemically stained slides, the expression area was analyzed using a slide scanner (Axio Scan Z1, Carl Zeiss, Germany), and the slides were then subjected to image analysis (ZEN, Carl Zeiss, Germany).

## Western blotting

Cryogenically frozen samples were homogenized using RIPA buffer, after which proteins were extracted and quantified with a protein assay kit (Bio-Rad, CA, USA). A sample for electrophoreses was prepared by quantifying the sample's protein content, and the sample was electrophoresed on 10–14% acrylamide gel for 120 minutes. The protein was then transferred to a polyvinylidene difluoride (PVDF) membrane, and non-specific protein binding sites were removed using blocking buffer. The PVDF membrane was incubated with interleukin-6 (IL-6), osteopontin (OPN), and bone morphogenetic protein 2 (BMP-2) primary antibodies at about 4˚C for 6 hours or more. After the primary antibody reaction, the membrane was exposed to the secondary antibody (diluted 1: 10,000). After the reaction was complete, the cells were washed with PBS-T buffer (0.5% Tween 20 in phosphate buffered saline) and the sample was developed using an enhanced chemiluminescence (ECL) reagent for immunoblot analysis. The color-completed sample was analyzed using an image analyzer. β-actin was used as intrinsic control. Finally, the ECL signal was quantitated using the pixel density analysis

algorithm within ImageJ software (National Institute of Health, NY, USA). The relative band density was calculated as follows: relative band density = (specific band density/β-actin band density) × 100.

## Statistical analysis

All data are expressed as mean ± SD. Statistical analyses were performed using GraphPad Prism 5.0 software (GraphPad Software, Inc., San Diego, CA, USA). An unpaired t-test was used to compare two groups. All statistical tests were two-sided, and significance was defined as $P < 0.05$.

## Results

There were no differences in body weight between the control and Entelon150®-treated groups (Fig 2A). Vascular patency was evaluated using ultrasonography and CT scanning. No interruption of flow patency was observed on either CT scans (Fig 2B and 2C) or ultrasonographs (Fig 3). Although there are no occlusions visible around the vessel or in the implants in either control or Entelon150®-treated groups, the vessels shown have an overall smaller and more irregular diameter might be for suturing effect (Fig 2). The $Ca^{2+}$ level in the Entelon150®-treated group (0.56±0.14 mg/g) was significantly lower than that in the NC group (1.48±0.57 mg/g) ($P < 0.05$, Fig 4). Western blot analysis showed that BMP-2 levels in the Entelon150®-treated group (82.21±11.20%) were significantly lower ($P < 0.001$) than in the NC group (100.00±4.63%). Western blot analysis also revealed that IL-6 levels in the Entelon150®-treated group (55.36±5.49%) were significantly lower ($P < 0.001$) than NC group (100.00±10.30%) indicating the significant attenuation of inflammation. However, the expression of OPN was not significantly different between two groups (Fig 5). Histopathological examination revealed infiltration of chronic inflammatory cells such as fibroblasts and macrophages around the graft in all groups. However, the inflammation level of the Entelon150®-treated group (1.50±0.58%) was significantly lower ($P < 0.001$) than the NC group (2.25±0.96%). In particular, a basophilic substance presumed to be the earliest sign of calcium deposition was observed in the NC group between the intercellular matrixes of the peri-graft tissue (Fig 6). Immunohistochemical staining revealed that BMP-2 levels in the Entelon150®-treated group (1.27±0.06%) were significantly ($P < 0.05$) lower than those in the NC group (1.67±0.31%). However, the expression of α-SMA was not significantly different between the two groups (Fig 7).

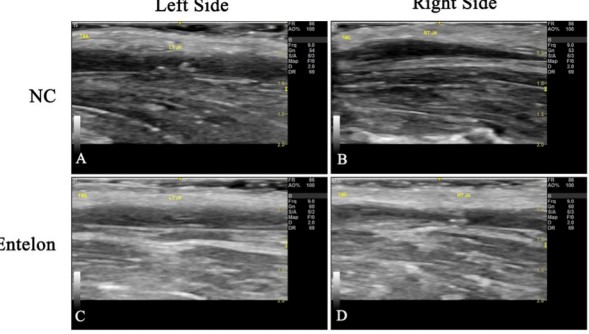

**Fig 3. Effects of bioprosthetic valve implantation and Entelon® treatment on vascular patency as measured by ultrasonography.** NC: Negative control group. Entelon®: Entelon®-treated group. The fluent vascular patency was confirmed by Ultrasonography".

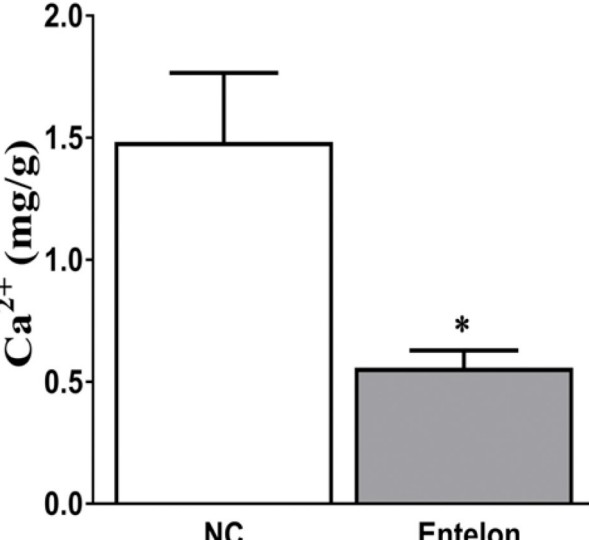

**Fig 4. Therapeutic efficacy of Entelon® treatment on the calcium content in bioprosthetic valve implants.** NC: Negative control group, Entelon®: Entelon®-treated group. The data are reported as mean ± SD. *: $p < 0.05$; and ***: $p < 0.001$, Bonferroni post hoc test following one-way ANOVA versus the NC group.

## Discussion

This study was the first to investigate the anti-inflammatory and anti-calcification effects of Entelon150® using a beagle dog model of intravenous bovine pericardium implantation. Chronic inflammation and calcification are major signs of the structural degeneration, dysfunction, and failure of bioprosthetic valve made of bovine pericardium [3, 14, 15]. We hypothesized that Entelon150® would reduce the expression of inflammatory cytokines in the vascular tissue of bovine pericardial implants, thereby suppressing calcification and bovine pericardium. In this study we demonstrated that Entelon150® treatment had two effects: significantly attenuating both inflammation and calcification in a beagle dog model of intravascular bovine pericardial implantation.

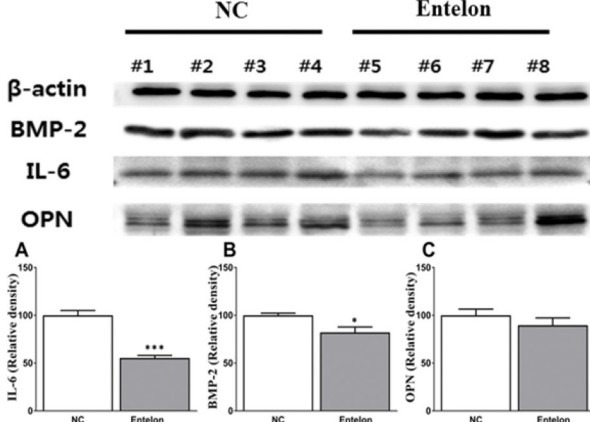

**Fig 5. Effects of bioprosthetic valve implantation and Entelon® treatment on the protein expression of BMP-2, IL-6 and OPN as analyzed by western blot.** β-Actin was used as an internal control. NC: Negative control group, Entelon®: Entelon®-treated group. IL-6: Interleukin-6, OPN: osteopontin, BMP-2: bone morphogenetic protein 2. The data are reported as mean ± SD. *: $p < 0.05$; and ***: $p < 0.001$, Bonferroni post hoc test following one-way ANOVA versus the NC group.

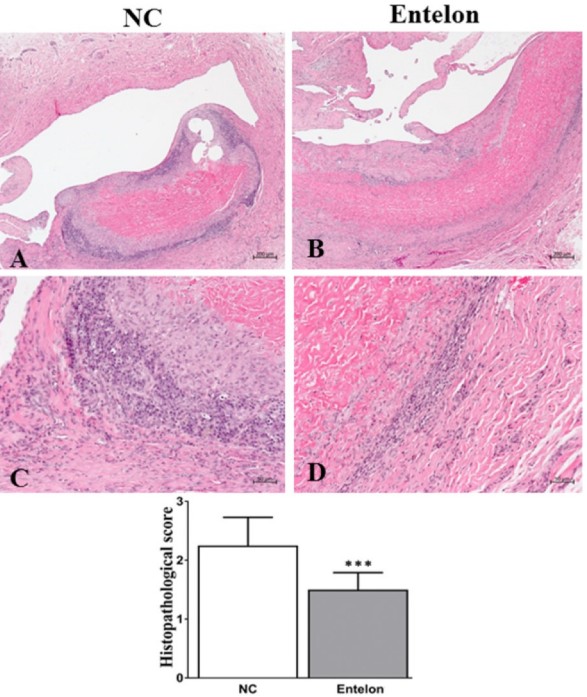

**Fig 6. Evaluation of the therapeutic efficacy of Entelon® using histological images analysis.** NC: Negative control group, Entelon®: Entelon®-treated group. The data are reported as mean ± SD. ***: p < 0.001, Bonferroni post hoc test following one-way ANOVA versus the NC group.

Calcification or degenerative changes of implant valve in human usually takes longtime [16]. However, degeneration was observed comparatively with in shorter time in experimental implant model; evaluated after 14 and 30 days [17], 4 and 12 weeks [14], 3 weeks [18], 6 weeks [3, 19, 29] implantation. Subcutaneous and intramuscular implantation models are usually performed to investigate BHV degenerative changes or calcification [14, 17–19]. Unfortunately, subcutaneous and intramuscular models are not appropriate as graft tissue experienced hemodynamic stress due to the direct blood contact or circulating factors inside the heart or great vessels. Therefore, intravascular implant model became popular model [3, 29]. We therefore performed intravascular implant model.

Indeed, immunological rejection of heterologous tissue is a challenging medical problem contributing to post-implant xenograft degeneration [19]. Our histopathological results are consistent with previous studies observing infiltration of chronic inflammatory cells such as fibroblasts and macrophages around the graft. The inflammation levels of the Entelon150®-treated group was significantly lower than the negative control group. In addition, a basophilic substance, presumed to be the earliest sign of calcium deposition, was observed between the intercellular matrixes of the peri-graft tissue in the negative control group. The basophilic staining using the H & E stain method indicated calcium deposition [20]. The results suggest that administration of Entelon150® lowered inflammation levels and inhibited calcium deposition in the tissues surrounding the graft.

In addition, we found that bovine pericardium triggered an immunological response, as we observed a significant elevation of IL-6 in the NC group which were significantly reduced by Entelon150® treated group. Steroidal anti-inflammatory therapy significantly reduces the incidence of postoperative valve tissue rejection in patients, indicating that suppressing the valve-induced immunological response may improve the postoperative durability of bioprosthetic

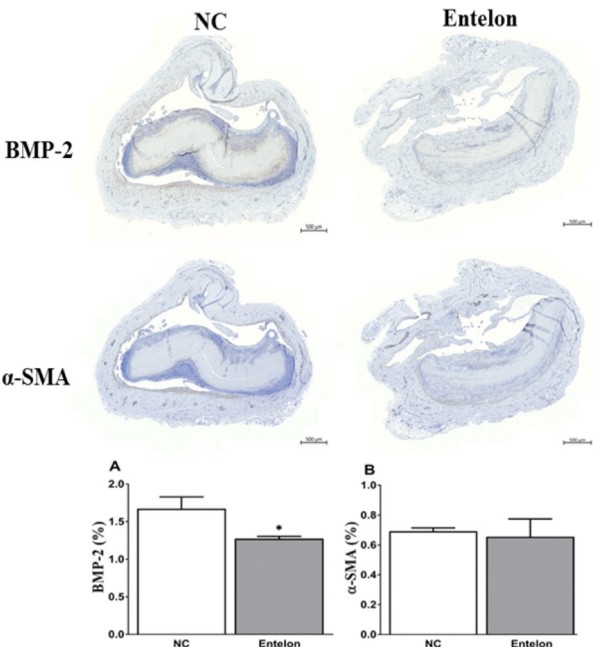

**Fig 7. Evaluation of the therapeutic efficacy of Entelon® using histological images analysis.** NC: Negative control group, Entelon®: Entelon®-treated group. The data are reported as mean ± SD. *: p < 0.05, Bonferroni post hoc test following one-way ANOVA versus the NC group.

aortic valve implants [21]. Importantly, our data showed that Entelon150® treatment significantly lowered IL-6 levels, thus mitigating inflammation.

BMP-2 is a member of the transforming growth factor (TGF) superfamily and is known to be a master regulator of conventional and ectopic osteogenesis [22]. Alteration of BMP-2 reportedly aggravates skeletal and extraskeletal mineralization [23]. In addition, BMP-2 plays a critical role in vascular disease, including atherosclerosis and plaque instability through its effects on vascular inflammation. BMP-2 also regulates vascular oxidative stress and vascular calcification by stimulating osteogenesis in vascular smooth muscle cells [24]. Furthermore, IL-6 activity is strongly associated BMP-2 expression [3, 14] and calcification [25]. In this study we found that calcium content and IL-6 expression were significantly lowered in the bovine pericardium of the Entelon150®-treated group. To elucidate the underlying mechanism involved, we evaluated BMP-2 expression in implanted tissue. Interestingly, Entelon150® treatment significantly lowered BMP-2 expression, demonstrating its therapeutic molecular effects. OPN is an extracellular matrix glycoprotein mainly taking part in bone morphogenesis, bio-mineralization and calcification. OPN is produced as a cytokine in activated T cells and macrophages, demonstrating that OPN plays an important role modulating inflammation. During the healing process or under pro-inflammatory conditions, OPN expression is elevated near inflammatory cells. OPN is reportedly associated with inflammation, atherosclerosis, and vascular calcification [26]. We also found that along with IL-6 and calcification, OPN expression was increased in the implanted tissue and was non significantly lowered by Entelon150® administration. Additionally, α-SMA expression is used as a measure of tissue fibrosis. Calcification and fibrosis have many common features such as risk factors and have histopathological lesions with similar pathogenic pathways and mediators. The factors initiating calcification include inflammation, cell injury, and tissue infiltration by inflammatory cells, lipids, cytokines, and reactive oxygen species and the overexpression of α-SMA in

calcified tissue has also been reported [27, 28]. Consistent with these findings, the expression α-SMA in the bovine pericardium was lowered by Entelon150® treatment; however, this difference was not statistically significant.

We have previously found that the angiotensin II type 1 receptor blocker losartan attenuates bioprosthetic valve leaflet calcification in a rabbit model of intravascular implantation [3]. Calcification of bovine pericardium appears unrelated to specific mechanisms, but rather appears related to reduced inflammation and substances like IL-6, BMP-2, and OPN. Any substance that lowers inflammation through IL-6, BMP-2, and OPN may help prevent calcification. From this point of view, Entelon150®, which consists of grape seed extract, will be more powerful than other synthetic medications at preventing calcification.

## Limitations

The limitations of this study are its relatively low number and the uncertain mechanism of degenerative calcification in our beagle dog model of intravascular bovine pericardium implantation. It is unclear whether the mechanisms in our model are similar to the mechanisms underlying calcification of bovine pericardium in humans. However, we previously compared five implantation methods in a rabbit model and found that the intravenous implantation model most closely resembled bioprosthetic valve made of bovine pericardium calcification in humans. Furthermore, we reported that the calcium content was higher in intravenous implants than in arterial patch implants [29]. We performed our experiments in beagle dogs rather than rabbits; however, we think our results are consistent with the findings in our rabbit intravascular model.

## Conclusion

We found that Entelon 150® significantly attenuated post-implant degenerative changes inflammation and calcium deposition in a beagle dog of intravascular bovine pericardium implantation model. Further observations are required to assess the effects of Entelon 150® on native vessel calcification in another animal model.

## Supporting information

**S1 Raw images.**
(PDF)

## Author Contributions

**Conceptualization:** Ji Hyun Oh, Yun Seok Cho, Hong Ju Shin.

**Data curation:** Sokho Kim, Yun Seok Cho.

**Investigation:** Sokho Kim, Hong Ju Shin.

**Resources:** Ji Hyun Oh.

**Writing – original draft:** Gab-Chol Choi, Sokho Kim, Md. Mahbubur Rahman, Ji Hyun Oh.

**Writing – review & editing:** Yun Seok Cho, Hong Ju Shin.

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
