## [Decision Letter · Decision Letter 0]

17 Nov 2020

PONE-D-20-18358

Entelon® (vitis vinifera seed extract) reduces inflammation and calcification in a beagle dog model of intravascular bovine pericardium implantation

PLOS ONE

Dear Dr. Shin,

Thank you for submitting your manuscript to PLOS ONE. After careful consideration, we feel that it has merit but does not fully meet PLOS ONE’s publication criteria as it currently stands. Therefore, we invite you to submit a revised version of the manuscript that addresses the points raised during the review process.

We look forward to receiving your revised manuscript.

Kind regards,

Andreas Zirlik, MD

Academic Editor

PLOS ONE

Journal Requirements:

2. At this time, we request that you  please report additional details in your Methods section regarding animal care, as per our editorial guidelines:

(1) Please state the source of the dogs used in the study  

(2) Please include the method of euthanasia

(3) Please describe the post-operative care received by the dogs, including the frequency of monitoring and the criteria used to assess animal health and well-being.  

Thank you for your attention to these requests.

3. We note that Figure 1 in your submission contains copyrighted images. All PLOS content is published under the Creative Commons Attribution License (CC BY 4.0), which means that the manuscript, images, and Supporting Information files will be freely available online, and any third party is permitted to access, download, copy, distribute, and use these materials in any way, even commercially, with proper attribution. For more information, see our copyright guidelines: http://journals.plos.org/plosone/s/licenses-and-copyright.

(1) You may seek permission from the original copyright holder of Figure 1 to publish the content specifically under the CC BY 4.0 license.

4. Please provide the source, product number and any lot numbers of the Entelon150 purchased from Hanlim Pharm. Co., Ltd  for your study.”

Reviewers' comments:

Reviewer's Responses to Questions

**Comments to the Author**

1. Is the manuscript technically sound, and do the data support the conclusions?

Reviewer #1: Partly

Reviewer #2: Yes

2. Has the statistical analysis been performed appropriately and rigorously? 

Reviewer #1: Yes

Reviewer #2: Yes

3. Have the authors made all data underlying the findings in their manuscript fully available?

Reviewer #1: No

Reviewer #2: Yes

4. Is the manuscript presented in an intelligible fashion and written in standard English?

Reviewer #1: Yes

Reviewer #2: Yes

5. Review Comments to the Author

Reviewer #1: Dear Editor!

Thank you for the opportunity to review this interesting paper! Choi and colleagues evaluated the local reaction regarding inflammation and calcification after bovine patch implantation and the potential modification with systemic treatment of Entelon over six weeks, which is a grape seed extract.

While the findings are interesting, I do have some concern regarding the applicability and interpretation of the results which needs to be addressed:

Major:

- The authors conclude that the systemic treatment reduced the calcification in the patch area. However, despite reduced calcium concentration, no direct calcification could be shown. This would also be very early after six weeks. Therefore, I would suggest to change the conclusion and drive the attention towards the changes is inflammatory reaction.

- In general, the concept of systemic treatment to hinder degeneration of an implanted bioprosthesis is interesting and needs further discussion. It seems obvious that for example full immunosuppression would avoid calcification of aortic or pulmonary native homograft implantation, as this is not seen after heart transplantation. However, nobody would treat a patient after homograft implantation with full immunosuppression due to the side effects of a life-long therapy. Therefore, the current approach is to avoid immunoreaction towards homografts by decellularization to avoid the immune reaction. If a systemic therapy needs to be administered after bovine valve implantation to hinder tissue calcification, this therapy requires to have almost no side effects over 10-20 years, which is the current typical life-time of a bioprosthetic valve. Therefore, the applied dose of Entelon and the associated expected risk profile in human administration needs to be addressed and discussed. Furthermore, alternative therapies like statins have previously been evaluated and needs to be addressed.

- The authors mention the low case number in the limitation session. This is indeed a critical issue. A very strong underlying effect is to be expected if significant differences were observed with 4 animals per group. A higher sample size would be beneficial.

- It seems interesting that Entelon is reducing BMP-2 concentration in the affected tissue as the potential underlying mechanism. Was this effect previously observed? What is the potential mechanism to change BMP-2 activity?

Minor:

- The patch was implanted in the venous position. However, an arterial patch would have more stress and a potential earlier degeneration. Why was a venous place chosen?

Reviewer #2: Dear Authors,

selecting the topic “Inflammation and calcification in the setting of bio-prosthetic valve and other substitute heart valve implantation failure, you are taking on an ongoing problem in current state-of-the-art medical therapy.

The authors are able to show that a substance known to reduce overall inflammatory signaling is able to reduce calcium content as well as the inflammation level in the specific setting of intravenous bovine pericardium implants in dogs.

The study itself is well-designed and overall, the manuscript is well written and presents the findings in a comprehensive manner. The figures present the clinical as well as in-vitro data in a concise way, while surely benefitting from some more details.

However, there are several points that need clarification which will surely benefit the manuscript and make it more easy to follow.

1. Line 260ff: The conclusion should be rephrased. In line 171, the authors mention there were no calcified lesions detectable in any of the animals. Also the follow up of only 6 weeks is rather short term whereas degenerative calcification is a long term process. On the other hand, the significant attenuation of inflammation is not mentioned

2. Line 112ff: When collecting the sample you describe storing the prosthetic — was this with or without the surrounding venous tissue? Please clarify where lesions and calcification were examined. Right now, it remains unclear if this analysis was performed on the same tissues.

3. Western Blotting:

Line 147: how was the protein content of samples quantified? Did you perform a Bradford Assay or some different method?

4. Line 153: I suspect you washed the membranes after staining with secondary antibody? Please correct or elaborate further on the procedure.

5. Line 167ff: Please specify the time point(s) at which body weight and vascular patency were examined in the results section. At the moment there is only 1 side note in the methods section under “sample collection”. Also, I can only assume the CT scan was performed at the same time as ultrasound examination took place? Please specify

6. Line 171: Please define “calcified lesion”. You state that there were no lesions in the implant in either group. However, overall Calcium content was higher in the negative control group. Are the measured Calcium levels after bovine patch implantation higher as compared to untreated veins?

7. Line 243: please specify which “certain mechanism” you are talking about. In its current form, this statement provides no value to the reader.

8. Line 246: It is unclear to me as to why a phytochemical substance should be more powerful than other systemic agents? You state that “any substance” lowering inflammation through a given signaling pathway may help prevent calcification. Please rephrase.

9. Figure 1:

Please label the anatomical landmarks mentioned in the manuscript and relevant for understanding figure 2 B+C. Also, it would be helpful to indicate the location of incision and subsequent bovine pericardium patch implantation into the schematic drawing to exclude any ambiguity.

10. Figure2:

Please label the vessels and anatomical structures shown (see comment for figure1). Also, although there are no occlusions visible, one has the impression that in the control-group, the vessels shown have an overall smaller and more irregular diameter. This should be discussed in the manuscript.

6. PLOS authors have the option to publish the peer review history of their article (what does this mean?). If published, this will include your full peer review and any attached files.

Reviewer #1: No

Reviewer #2: No

---

## [Author Response · Author response to Decision Letter 0]

14 Jan 2021

2. At this time, we request that you please report additional details in your Methods section regarding animal care, as per our editorial guidelines:

Please state the source of the dogs used in the study

⇒ According to this comment, I deleted the latter “multiple The source of dogs is added, sorry for this mistake

“Eight healthy male beagle dogs (20 weeks old; mean body weight 9.76 ± 0.32 kg, range 8.00–10.80 kg) were used in this study which were purchased from ORIENT BIO Inc. (Seongnam, Gyeonggi-do, Republic of Korea)”.

(2) Please include the method of euthanasia

⇒ According to this comment, Euthenesia methods are added. We are sorry for this mistake- 

 “Six weeks after surgery and immediately after measuring vascular patency, the animals were euthanized by propofol (10 mg/kg, IV) and pentobarbital (50 mg/kg, IV)”. 

(3) Please describe the post-operative care received by the dogs, including the frequency of monitoring and the criteria used to assess animal health and well-being. 

⇒ Dear respected editor, According to your comment postoperative care and monitoring are included details 

“Postoperatively, intramuscular enrofloxacine (10 mg/kg, bid), intravenous cephradine (30 mg/kg bid or tid), intravenous tramadol (2–3 mg/kg bid or tid) and intravenous cimetidine (10 mg/kg, bid) were performed for three days [12]. Respiratory rate was monitored by the movement of the abdominal and chest wall (breath/min) twice daily. Heart rate and ECG was recorded with a Cardiofax ECG-9020 electrocardiograph (Nihon Kohden, Tokyo, Japan) once daily for three days [13]. Beside these normal clinical sings were closely monitored daily until end of the experiment to identify any abnormality due to surgery or any side effect for administering Entelon150 including feeding and drinking behavior, urination and defecation frequency, bleeding, salivating, vomiting, abnormality or redness of skin and eye color, abnormal sound and abnormal movement but none were observed in entire experimental period.”.

3. We note that Figure 1 in your submission contains copyrighted images. All PLOS content is published under the Creative Commons Attribution License (CC BY 4.0), which means that the manuscript, images, and Supporting Information files will be freely available online, and any third party is permitted to access, download, copy, distribute, and use these materials in any way, even commercially, with proper attribution. For more information, see our copyright guidelines: http://journals.plos.org/plosone/s/licenses-and-copyright.

(1) You may seek permission from the original copyright holder of Figure 1 to publish the content specifically under the CC BY 4.0 license.

⇒ Dear respected editor, Fig. 1 is our original drawing. When we submited this manuscript in PLOS ONE first, there was asking to reveal our data previously in bioRxiv. Thus, we permitted it. According to bioRxiv preprint doi: https://doi.org/10.1101/2020.06.17.156695; this version posted June 17, 2020. The copyright holder for this preprint is the author/funder. who has granted bioRxiv a license to display the preprint in perpetuity. It is made available under aCC-BY 4.0 International license. And the corresponding author is Hong Ju Shin. Fig. 1 is important picture of this manuscript. Dr. Shin made this picture with illustrator at first. 

4. Please provide the source, product number and any lot numbers of the Entelon150 purchased from Hanlim Pharm. Co., Ltd for your study.”

⇒ The source and lot number is provided in material and methods section

“(Entelon150®; Lot number: RNR601, Hanlim Pharm. Co., Ltd. Yongin-si, Korea)”.

Reviewers' comments:

Reviewer's Responses to Questions

Comments to the Author

1. Is the manuscript technically sound, and do the data support the conclusions?

Reviewer #1: Partly

Reviewer #2: Yes

2. Has the statistical analysis been performed appropriately and rigorously?

Reviewer #1: Yes

Reviewer #2: Yes

3. Have the authors made all data underlying the findings in their manuscript fully available?

Reviewer #1: No

Reviewer #2: Yes

4. Is the manuscript presented in an intelligible fashion and written in standard English?

Reviewer #1: Yes

Reviewer #2: Yes

5. Review Comments to the Author Please use the space provided to explain your answers to the questions above. You may also include additional comments for the author, including concerns about dual publication, research ethics, or publication ethics. (Please upload your review as an attachment if it exceeds 20,000 characters)

Reviewer #1: 

Dear Editor! Thank you for the opportunity to review this interesting paper! Choi and colleagues evaluated the local reaction regarding inflammation and calcification after bovine patch implantation and the potential modification with systemic treatment of Entelon over six weeks, which is a grape seed extract. While the findings are interesting, I do have some concern regarding the applicability and interpretation of the results which needs to be addressed:

Major:

The authors conclude that the systemic treatment reduced the calcification in the patch area. However, despite reduced calcium concentration, no direct calcification could be shown. This would also be very early after six weeks. Therefore, I would suggest to change the conclusion and drive the attention towards the changes is inflammatory reaction.

- In general, the concept of systemic treatment to hinder degeneration of an implanted bioprosthesis is interesting and needs further discussion. It seems obvious that for example full immunosuppression would avoid calcification of aortic or pulmonary native homograft implantation, as this is not seen after heart transplantation. However, nobody would treat a patient after homograft implantation with full immunosuppression due to the side effects of a life-long therapy. Therefore, the current approach is to avoid immunoreaction towards homografts by decellularization to avoid the immune reaction. If a systemic therapy needs to be administered after bovine valve implantation to hinder tissue calcification, this therapy requires to have almost no side effects over 10-20 years, which is the current typical life-time of a bioprosthetic valve. Therefore, the applied dose of Entelon and the associated expected risk profile in human administration needs to be addressed and discussed. Furthermore, alternative therapies like statins have previously been evaluated and needs to be addressed. 

⇒ After consideration your comments we have changes “Title”, and rewrite “Discussion” and “Conclusion”. 

Title: “Entelon® (vitis vinifera seed extract) reduces degenerative changes in bovine pericardium valve leaflet in a dog intravascular implant model”

Discussion:

“Calcification or degenerative changes of implant valve in human usually takes longtime [16]. However, degeneration was observed comparatively with in shorter time in experimental implant model; evaluated after 14 and 30 days [17], 4 and 12 weeks [14], 3 weeks [18], 6 weeks [3,19, 29] implantation. Subcutaneous and intramuscular implantation models are usually performed to investigate BHV degenerative changes or calcification (14,17,18,19). Unfortunately, subcutaneous and intramuscular models are not appropriate as graft tissue experienced hemodynamic stress due to the direct blood contact or circulating factors inside the heart or great vessels. Therefore, intravascular implant model became popular model [3,29]. We therefore performed intravascular implant model”.

Conclusion:

“We found that Entelon 150® significantly attenuated post-implant degenerative changes inflammation and calcification in a beagle dog of intravascular bovine pericardium implantation model. Further observations are required to assess the effects of Entelon 150® on native vessel calcification in another animal model”.

- The authors mention the low case number in the limitation session. This is indeed a critical issue. A very strong underlying effect is to be expected if significant differences were observed with 4 animals per group. A higher sample size would be beneficial.

⇒ Dear respected reviewer we also mentioned it in our limitation part 

- It seems interesting that Entelon is reducing BMP-2 concentration in the affected tissue as the potential underlying mechanism. Was this effect previously observed? What is the potential mechanism to change BMP-2 activity?

⇒ Dear respected reviewer we have mentioned the underlying mechanism and relationship of BMP-2 with inflammation and calcification in “Discussion”-

 “BMP-2 is a member of the transforming growth factor (TGF) superfamily and is known to be a master regulator of conventional and ectopic osteogenesis [17]. Alteration of BMP-2 reportedly aggravates skeletal and extraskeletal mineralization [18]. In addition, BMP-2 plays a critical role in vascular disease, including atherosclerosis and plaque instability through its effects on vascular inflammation. BMP-2 also regulates vascular oxidative stress and vascular calcification by stimulating osteogenesis in vascular smooth muscle cells [19]. Furthermore, IL-6 activity is strongly associated BMP-2 expression [3,12] and calcification [20]. In this study we found that calcium content and IL-6 expression were significantly lowered in the bovine pericardium of the Entelon150®-treated group. To elucidate the underlying mechanism involved, we evaluated BMP-2 expression in implanted tissue. Interestingly, Entelon150® treatment significantly lowered BMP-2 expression, demonstrating its therapeutic molecular effects.

Minor:

The patch was implanted in the venous position. However, an arterial patch would have more stress and a potential earlier degeneration. Why was a venous place chosen?

⇒ Dear respected reviewer, we previously compared five implantation methods in a rabbit model and found that the calcium content was higher in intravenous implants than in arterial patch implants (J Heart Valve Dis 2015;24:621-8). Actually it is technically difficult to implant intra-arterial patch in small carotid artery (2~3mm size). Therefore, we chose intravenous implants model. 

Reviewer #2:

Dear Authors, selecting the topic “Inflammation and calcification in the setting of bio-prosthetic valve and other substitute heart valve implantation failure, you are taking on an ongoing problem in current state-of-the-art medical therapy. The authors are able to show that a substance known to reduce overall inflammatory signaling is able to reduce calcium content as well as the inflammation level in the specific setting of intravenous bovine pericardium implants in dogs.

The study itself is well-designed and overall, the manuscript is well written and presents the findings in a comprehensive manner. The figures present the clinical as well as in-vitro data in a concise way, while surely benefitting from some more details. However, there are several points that need clarification which will surely benefit the manuscript and make it more easy to follow.

1. Line 260ff: The conclusion should be rephrased. 

⇒ We have rephrased conclusion. Could you please see below -

Conclusion:

“We found that Entelon 150® significantly attenuated post-implant degenerative changes inflammation and calcium deposition in a beagle dog of intravascular bovine pericardium implantation model. Further observations are required to assess the effects of Entelon 150® on native vessel calcification in another animal model”.

In line 171, the authors mention there were no calcified lesions detectable in any of the animals. 

⇒ Dear respected reviewer it was our mistake; we would like to say any type of occlusion was not observed. We have corrected the sentence. Could you please see below -

“Although there are no occlusions visible around the vessel or in the implants in either control or Entelon150®-treated groups, the vessels shown have an overall smaller and more irregular diameter might be for suturing effect (Fig.2)”.

Also the follow up of only 6 weeks is rather short term whereas degenerative calcification is a long term process. 

⇒ Thank you for your comments. As you commented, We changed “Title”, and rewrite 

“Discussion” and “Conclusion”. 

Title: “Entelon® (vitis vinifera seed extract) reduces degenerative changes in bovine pericardium valve leaflet in a dog intravascular implant model”

Discussion:

“Calcification or degenerative changes of implant valve in human usually takes longtime [16]. However, degeneration was observed comparatively with in shorter time in experimental implant model; evaluated after 14 and 30 days [17], 4 and 12 weeks [14], 3 weeks [18], 6 weeks [3,19, 29] implantation. Subcutaneous and intramuscular implantation models are usually performed to investigate BHV degenerative changes or calcification (14,17,18,19). Unfortunately, subcutaneous and intramuscular models are not appropriate as graft tissue experienced hemodynamic stress due to the direct blood contact or circulating factors inside the heart or great vessels. Therefore, intravascular implant model became popular model [3,29]. We therefore performed intravascular implant model”.

Conclusion:

“We found that Entelon 150® significantly attenuated post-implant degenerative changes inflammation and calcification in a beagle dog of intravascular bovine pericardium implantation model. Further observations are required to assess the effects of Entelon 150® on native vessel calcification in another animal model”.

On the other hand, the significant attenuation of inflammation is not mentioned 

⇒ Dear respected reviewer we have now corrected and mentioned in several times that treatment reduced inflammation. Could you please see below -

Result:

“Western blot analysis also revealed that IL-6 levels in the Entelon150®-treated group (55.36±5.49%) were significantly lower (P < 0.001) than NC group (100.00±10.30%) indicating the significant attenuation of inflammation”.

“Histopathological examination revealed infiltration of chronic inflammatory cells such as fibroblasts and macrophages around the graft in all groups. However, the inflammation level of the Entelon150®-treated group (1.50±0.58%) was significantly lower (P < 0.001) than the NC group (2.25±0.96%). In particular, a basophilic substance presumed to be the earliest sign of calcium deposition was observed in the NC group between the intercellular matrixes of the peri-graft tissue (Fig 6)”.

Discussion: 

“In this study we demonstrated that Entelon150® treatment had two effects: significantly attenuating both inflammation and calcification in a beagle dog model of intravascular bovine pericardial implantation”.

“Indeed, immunological rejection of heterologous tissue is a challenging medical problem contributing to post-implant xenograft degeneration [1914]. Our histopathological results are consistent with previous studies observing infiltration of chronic inflammatory cells such as fibroblasts and macrophages around the graft. The inflammation levels of the Entelon150®-treated group was significantly lower than the negative control group. In addition, a basophilic substance, presumed to be the earliest sign of calcium deposition, was observed between the intercellular matrixes of the peri-graft tissue in the negative control group. The basophilic staining using the H & E stain method indicated calcium deposition [2015]. The results suggest that administration of Entelon150® lowered inflammation levels and inhibited calcium deposition in the tissues surrounding the graft”.

“In addition, we found that bovine pericardium triggered an immunological response, as we observed a significant elevation of IL-6 in the NC group which were significantly reduced by Entelon150® treated group. Steroidal anti-inflammatory therapy significantly reduces the incidence of postoperative valve tissue rejection in patients, indicating that suppressing the valve-induced immunological response may improve the postoperative durability of bioprosthetic aortic valve implants [2116]. Importantly, our data showed that Entelon150® treatment significantly lowered IL-6 levels, thus mitigating inflammation.”

2. Line 112ff: When collecting the sample, you describe storing the prosthetic — was this with or without the surrounding venous tissue? Please clarify where lesions and calcification were examined. Right now, it remains unclear if this analysis was performed on the same tissues.

⇒ Dear respected reviewer we rewrote the according to your nice comment-

 “Only implanted prosthetic was collected from the right external jugular vein and connective tissues around the implanted prosthetic was carefully removed and then stored in a cryogenic freezer maintained at about -80 ° C until calcium quantification. However, half of the left external jugular vein along with connective tissue was fixed in a 10% neutral buffered formalin solution for histopathological examination, and from the other half part only prosthetic was stored in a cryogenic freezer maintained at -80 ° C for western blotting”.

- 3. Western Blotting: Line 147: how was the protein content of samples quantified? Did you perform a Bradford Assay or some different method?

⇒ We have added the assay method. Sorry for this mistake

 “Cryogenically frozen samples were homogenized using RIPA buffer, after which proteins were extracted and quantified with a protein assay kit (Bio-Rad, CA, USA)”.

4. Line 153: I suspect you washed the membranes after staining with secondary antibody? Please correct or elaborate further on the procedure.

⇒ Dear respected reviewer we have already mentioned it. Could you please see below-

 “After the primary antibody reaction, the membrane was exposed to the secondary antibody (diluted 1: 10,000). After the reaction was complete, the cells were washed with PBS-T buffer (0.5% Tween 20 in phosphate buffered saline) and the sample was developed using an enhanced chemiluminescence (ECL) reagent for immunoblot analysis”.

- 5. Line 167ff: Please specify the time point(s) at which body weight and vascular patency were examined in the results section. At the moment there is only 1 side note in the methods section under “sample collection”. 

Also, I can only assume the CT scan was performed at the same time as ultrasound examination took place? Please specify

⇒ Dear respected reviewer by following your nice comment we have mentioned the time schedule of body weight measurement, CT-scan, angiography evaluation. Could you please see below –

Time schedule of Body weight measurement

“Body weight was measured just before starting experiment and then once a week until six weeks (at o, 1, 2, 3, 4, 5 and 6 weeks) (Fig 2)”.

Time schedule of CT-scan and angiography evaluation

 “Vascular patency was checked using the color doppler mode of an ultrasonic device (LOGIQ e Ultrasound; GE Healthcare, Fairfield, USA) at 4 weeks and 6 weeks after the bioprosthetic implantation. Angiography was also performed with a bolus intravenous injection of 2 ml/kg iohexol (Omnipaque™, 300 mg I/ml; GE Healthcare) using a CT-scanner (CT; 16-channel multidetector; Bright Speed Elite, GE Healthcare, Fairfield, CT, USA) for additional confirmation just after ultrasonograpy and immediately before sacrifice at 6 weeks”.

6. Line 171: Please define “calcified lesion”. You state that there were no lesions in the implant in either group. However, overall Calcium content was higher in the negative control group. Are the measured Calcium levels after bovine patch implantation higher as compared to untreated veins?

⇒ Dear respected reviewer it was our mistake; we would like to say any type of occlusion was not observed. We have corrected the sentence. Could you please see below –

“Although there are no occlusions visible around the vessel or in the implants in either control or Entelon150®-treated groups, the vessels shown have an overall smaller and more irregular diameter might be for suturing effect (Fig.2)”. 

7. Line 243: please specify which “certain mechanism” you are talking about. In its current form, this statement provides no value to the reader.

⇒ Dear respected reviewer, I meant that calcification is not related to specific mechanism, but related to lower inflammation including IL-6,BMP-2, Ostenopontin.

Therefore, I switched certain to specific.

We have discussed with the references the underlying mechanism and relationship of IL6, BMP-2 and OPN with inflammation and calcification in “Discussion”-

“IL-6:

In addition, we found that bovine pericardium triggered an immunological response, as we observed a significant elevation of IL-6 in the NC group. Steroidal anti-inflammatory therapy significantly reduces the incidence of postoperative valve tissue rejection in patients, indicating that suppressing the valve-induced immunological response may improve the postoperative durability of bioprosthetic aortic valve implants [21]. Importantly, our data showed that Entelon150® treatment significantly lowered IL-6 levels, thus mitigating inflammation. 

BMP-2:

BMP-2 is a member of the transforming growth factor (TGF) superfamily and is known to be a master regulator of conventional and ectopic osteogenesis [22]. Alteration of BMP-2 reportedly aggravates skeletal and extraskeletal mineralization [23]. In addition, BMP-2 plays a critical role in vascular disease, including atherosclerosis and plaque instability through its effects on vascular inflammation. BMP-2 also regulates vascular oxidative stress and vascular calcification by stimulating osteogenesis in vascular smooth muscle cells [24]. Furthermore, IL-6 activity is strongly associated BMP-2 expression [3,14] and calcification [25]. In this study we found that calcium content and IL-6 expression were significantly lowered in the bovine pericardium of the Entelon150®-treated group. To elucidate the underlying mechanism involved, we evaluated BMP-2 expression in implanted tissue. Interestingly, Entelon150® treatment significantly lowered BMP-2 expression, demonstrating its therapeutic molecular effects. OPN:

OPN is an extracellular matrix glycoprotein mainly taking part in bone morphogenesis, bio-mineralization and calcification. OPN is produced as a cytokine in activated T cells and macrophages, demonstrating that OPN plays an important role modulating inflammation. During the healing process or under pro-inflammatory conditions, OPN expression is elevated near inflammatory cells. OPN is reportedly associated with inflammation, atherosclerosis, and vascular calcification [26]. We also found that along with IL-6 and calcification, OPN expression was increased in the implanted tissue and was non significantly lowered by Entelon150® administration. 

α-SMA:

Additionally, α-SMA expression is used as a measure of tissue fibrosis. Calcification and fibrosis have many common features such as risk factors and have histopathological lesions with similar pathogenic pathways and mediators. The factors initiating calcification include inflammation, cell injury, and tissue infiltration by inflammatory cells, lipids, cytokines, and reactive oxygen species and the overexpression of α-SMA in calcified tissue has also been reported [27,28]. Consistent with these findings, the expression α-SMA in the bovine pericardium was lowered by Entelon150® treatment; however, this difference was not statistically significant”.

8. Line 246: It is unclear to me as to why a phytochemical substance should be more powerful than other systemic agents? You state that “any substance” lowering inflammation through a given signaling pathway may help prevent calcification. Please rephrase.

⇒ Dear respected reviewer we mentioned it in the “Introduction part”. Could you please see below –

“Several approaches to reduce calcification have been attempted, including systemic anti-calcification agent administration. However, many of these approaches have been either ineffective or have produced unwanted side effects [3]. To overcome this drawback, studies on the complementary treatment of calcification have focused on traditional herbal medicines recently [4-6]”.

9. Figure 1: Please label the anatomical landmarks mentioned in the manuscript and relevant for understanding figure 2 B+C. Also, it would be helpful to indicate the location of incision and subsequent bovine pericardium patch implantation into the schematic drawing to exclude any ambiguity.

⇒ Dear respected reviewer we have corrected the figure according to your suggestion .

Fig1. Anatomical diagram of a beagle’s jugular vein during the implantation procedure.

Yellow star indicates the incision site of jugular vein

10. Figure2: Please label the vessels and anatomical structures shown (see comment for figure1). Also, although there are no occlusions visible, one has the impression that in the control-group, the vessels shown have an overall smaller and more irregular diameter. This should be discussed in the manuscript.

⇒ Dear respected reviewer we corrected figure-2 and we mentioned it in the “Result sectiont”. Could you please see below –

“Although there are no occlusions visible around the vessel or in the implants in either control or Entelon150®-treated groups, the vessels shown have an overall smaller and more irregular diameter might be suturing effect (Fig.2)”.

Fig 2. Effects of bioprosthetic valve implantation and Entelon® treatment on body weight and vascular patency as measured by CT-Scan.

Arrow indicates the incision and implanted site in jugular vein.

6. PLOS authors have the option to publish the peer review history of their article (what does this mean?). If published, this will include your full peer review and any attached files.

Do you want your identity to be public for this peer review? For information about this choice, including consent withdrawal, please see our Privacy Policy.

Reviewer #1: No

Reviewer #2: No

---

## [Decision Letter · Decision Letter 1]

11 Feb 2021

Entelon® (vitis vinifera seed extract) reduces degenerative changes in bovine pericardium valve leaflet in a dog intravascular implant model

PONE-D-20-18358R1

Dear Dr. Shin,

We’re pleased to inform you that your manuscript has been judged scientifically suitable for publication and will be formally accepted for publication once it meets all outstanding technical requirements.

Kind regards,

Andreas Zirlik, MD

Academic Editor

PLOS ONE

Additional Editor Comments (optional):

Reviewers' comments:

Reviewer's Responses to Questions

**Comments to the Author**

1. If the authors have adequately addressed your comments raised in a previous round of review and you feel that this manuscript is now acceptable for publication, you may indicate that here to bypass the “Comments to the Author” section, enter your conflict of interest statement in the “Confidential to Editor” section, and submit your "Accept" recommendation.

Reviewer #2: All comments have been addressed

2. Is the manuscript technically sound, and do the data support the conclusions?

Reviewer #2: Yes

3. Has the statistical analysis been performed appropriately and rigorously? 

Reviewer #2: I Don't Know

4. Have the authors made all data underlying the findings in their manuscript fully available?

Reviewer #2: Yes

5. Is the manuscript presented in an intelligible fashion and written in standard English?

Reviewer #2: Yes

6. Review Comments to the Author

Reviewer #2: Dear Authors,

thank you again for this study on Inflammation and calcification in the setting of bio-prosthetic valve and other substitute heart valve implantation failure, which is an ongoing problem in current state-of-the-art medical therapy

All of my points of concern raised in the initial submit were dealt with thoroughly.

Revising your manuscript has clearly added to the readability, putting your significant findings into focus and better clinical context.

Also, the revised figures now help in visualizing the implantation site and make the figures more consistent.

This certainly made your manuscript even better suited for practicing cardio-thoracic surgeons and cardiologists.

7. PLOS authors have the option to publish the peer review history of their article (what does this mean?). If published, this will include your full peer review and any attached files.

Reviewer #2: No

---

## [Editor Report · Acceptance letter]

23 Feb 2021

PONE-D-20-18358R1 

Entelon (*vitis vinifera* seed extract) reduces degenerative changes in bovine pericardium valve leaflet in a dog intravascular implant model 

Dear Dr. Shin:

I'm pleased to inform you that your manuscript has been deemed suitable for publication in PLOS ONE. Congratulations! Your manuscript is now with our production department. 

Kind regards, 

on behalf of

Univ. Prof. Dr. Andreas Zirlik 

Academic Editor

PLOS ONE